# Spatio-temporal variability of eDNA signal and its implication for fish monitoring in lakes

Alix Hervé[1,2,3], Isabelle Domaizon[2,4], Jean-Marc Baudoin[2,5], Tony Dejean[1], Pierre Gibert[2,3], Pauline Jean[1], Tiphaine Peroux[2,3], Jean-Claude Raymond[2,6], Alice Valentini[1], Marine Vautier[2,4], Maxime Logez[2,3,7] *

1 SPYGEN, Le Bourget du Lac, France, 2 Pole R&D ECLA, Le Bourget-du-Lac, France, 3 INRAE, Aix Marseille Université, RECOVER, Aix-en-Provence, France, 4 INRAE, UMR CARRTEL, Thonon-les-Bains, France, 5 OFB, Direction de la Recherche et de l'Appui Scientifique, Route Cézanne, Aix-en-Provence, France, 6 OFB, DR AURA, Thonon-les-Bains, France, 7 INRAE, UR RIVERLY, Villeurbanne, France

* maxime.logez@inrae.fr

**Data Availability Statement:** The data underlying the results presented in the study are available from figshare (DOI: 10.6084/m9.figshare. 20173217).

## Abstract

Environmental DNA (eDNA) metabarcoding is revolutionizing the monitoring of aquatic biodiversity. The use of eDNA has the potential to enable non-invasive, cost-effective, time-efficient and high-sensitivity monitoring of fish assemblages. Although the capacity of eDNA metabarcoding to describe fish assemblages is recognised, research efforts are still needed to better assess the spatial and temporal variability of the eDNA signal and to ultimately design an optimal sampling strategy for eDNA monitoring. In this context, we sampled three different lakes (a dam reservoir, a shallow eutrophic lake and a deep oligotrophic lake) every 6 weeks for 1 year. We performed four types of sampling for each lake (integrative sampling of sub-surface water along transects on the left shore, the right shore and above the deepest zone, and point sampling in deeper layers near the lake bottom) to explore the spatial variability of the eDNA signal at the lake scale over a period of 1 year. A metabarcoding approach was applied to analyse the 92 eDNA samples in order to obtain fish species inventories which were compared with traditional fish monitoring methods (standardized gillnet samplings). Several species known to be present in these lakes were only detected by eDNA, confirming the higher sensitivity of this technique in comparison with gillnetting. The eDNA signal varied spatially, with shoreline samples being richer in species than the other samples. Furthermore, deep-water samplings appeared to be non-relevant for regularly mixed lakes, where the eDNA signal was homogeneously distributed. These results also demonstrate a clear temporal variability of the eDNA signal that seems to be related to species phenology, with most of the species detected in spring during the spawning period on shores, but also a peak of detection in winter for salmonid and coregonid species during their reproduction period. These results contribute to our understanding of the spatio-temporal distribution of eDNA in lakes and allow us to provide methodological recommendations regarding where and when to sample eDNA for fish monitoring in lakes.

**Funding:** This study was supported by the French Biodiversity Agency (OFB), the French National Research Institute for Agriculture, Food and Environment (INRAE), the company SPYGEN and the French National Agency for Research and Technology (ANRT).

**Competing interests:** Teleo primers and the use of the amplified fragment for identifying fish species from environmental samples are patented by the CNRS and the Université Grenoble Alpes. This patent only restricts commercial applications and has no implications for the use of this method by academic researchers. SPYGEN owns a licence for this patent. A. H., T. D., P. J. and A.V. are research scientists at a private company specialising in the use of eDNA for species detection.

## Introduction

Aquatic ecosystems are among the most threatened systems worldwide [1, 2], facing a constant intensification of anthropogenic pressures that modify their functioning, affect their biota and ultimately the services they can provide [3] (https://cices.eu/). Environmental policies were established to protect continental aquatic ecosystems, such as the Water Framework Directive (WFD) in Europe (2000). Such policies rely on biomonitoring programmes that use standardised protocols to obtain reliable data comparable in space and time in order to assess the ecological status of waterbodies [4, 5] and their trajectories. Standardised protocols specifying each aspect of the sampling method to be used were therefore formalised. The CEN protocol was designed to sample lacustrine fish using multi-mesh gillnets, with protocol specificities depending on the lake surface area and precise recommendations for sampling periods [6]. This protocol made it possible to compare fish communities from diverse lakes, at a large spatial scale, in order to better understand the ecological patterns behind these communities and to obtain standardised data on which to base multimetric indices [4]. Several drawbacks, however, are associated with the use of multi-mesh gillnets: (a) this method is time-consuming and thus expensive, especially for the largest systems; (b) even with an appropriate sampling effort, some species are not caught easily because of the selectivity of gillnets that varies according to species ecology, size and abundance of taxa (i.e. pelagic species are more likely to be caught; abundant species are more easily captured; medium-sized individuals are more likely to be caught and kept in gillnets) [7]; (c) this method is also very invasive, leading to a high biological cost [8]. Consequently, alternative methods (e.g. echo-sounder techniques [8]) have been investigated to monitor fish biodiversity more efficiently and to promote non-invasive approaches.

Environmental DNA (eDNA)-based methods are part of these non-invasive alternative techniques and have been largely developed over the past decade to answer various ecological questions [9–12]. eDNA methods dealing with the detection of macro-organisms are based on collecting the DNA released by these organisms (e.g. mucus, gametes, skin flakes, etc.) in diverse environmental matrices [12, 13] such as soil [14], air [15–17] or water [18]. The first use of eDNA focused on monospecific approaches [12], to detect rare indigenous species [19, 20] or for the early detection of invasive species [21–23]. Improvements in sequencing methods with high-throughput sequencers made it possible to promote the use of metabarcoding approaches [24, 25], which represent an efficient tool for revealing the taxonomic composition of local biological assemblages of various phylogenetic groups, from bacteria to eukaryotic species [15, 26]. The eDNA metabarcoding workflow, from sampling to the final taxonomic list of species, is based on several steps with no international standardisation and with a diversity of protocols (i.e. sampling [27, 28], DNA extraction, amplification [29], library preparation, sequencing, bioinformatics treatments [30] and reference database establishment). Despite this diversity of protocols [31–33], eDNA methods have been successfully used to explore the distinct aspects of fish biodiversity, species occupancy [34] or density/biomass of targeted species [25].

The number of species that can be detected with eDNA is generally greater than with traditional methods [33, 35–37] providing a reliable assessment of fish communities [38–40]. However, our knowledge of the spatio-temporal variability of the eDNA signal is still incomplete. In lentic systems, eDNA spatial repartition appeared to be distributed unequally horizontally and vertically, due to biotic factors such as species distribution [18, 41, 42], behaviour for nutrition [43] or reproduction [44], but also due to abiotic conditions such as thermal stratification [39, 45, 46]. The spatial heterogeneity of the eDNA signal needs to be better understood and taken into account in order to design an efficient sampling strategy. It is also necessary to

assess the temporal dynamics of the spatial distribution of eDNA at lake scale. The challenge is to define where and when to sample so as to achieve an exhaustive assessment of fish biodiversity in order to formalise an operational protocol adapted to different typologies of lakes, from large deep lakes to small ecosystems. Some studies of lacustrine fish focused on these questions and considered either the spatial or the temporal variability of the eDNA signal [39, 47, 48], but rarely did they consider both of the two aspects. To our knowledge, these two factors (i.e. spatial and temporal variability) were analysed together only in three types of water bodies: an experimental pond [18], a natural lake [45] and a small dam reservoir [49]. This issue was never addressed through a comparative study of the spatio-temporal variability of eDNA for several types of lakes and considering possible changes over a year, to cover a complete annual cycle.

In order to improve our knowledge on the spatio-temporal variability of the fish eDNA signal over diverse lentic systems, we considered three different types of lakes in this study: a natural deep peri-alpine lake, a large dam reservoir and a small natural shallow lowland lake. We conducted eDNA sampling in these lakes, with four sampling strategies (sub-surface water collected with integrative sampling on the right shore, on the left shore, and at the centre of the lake; deeper waters collected by repeated punctual sampling in the bottom layers of the lake). The comparison of these sampling strategies aims at evaluating the potential spatial variability of the fish eDNA signal at the lake scale, and how this spatial distribution varies over an annual cycle in relation to the lake dynamics (i.e. thermal stratification and mixing) or the biological cycle of species (reproduction, hatching, etc.). Our objectives were to: (a) test the efficiency of eDNA metabarcoding in comparison with conventional methods of monitoring (here CEN protocol gillnets), (b) confirm that the eDNA method makes it possible to discriminate fish communities of the three water bodies, (c) compare the spatial distribution of the eDNA signal within lakes, and (d) reveal the potential temporal variability of the eDNA signal over the year. The final aim of this study was to provide recommendations concerning the sampling strategy (where and when to sample) that could be adopted to monitor fish in lakes using DNA-based methods.

## Materials and methods

### Study sites and context

The spatio-temporal survey was performed on three French lakes: a natural alpine lake, Lake Aiguebelette (45°33'30.24"N, 5°48'3.6"E), a large dam reservoir, Lake Serre-Ponçon (44° 30'52.92"N, 6°20'31.2"E), and a small lowland lake, Etang des Aulnes (43°35'30.84"N, 4° 47'33"E) (Fig 1).

Lake Aiguebelette is a natural, deep peri-alpine lake, located at an altitude of 374 m. This oligotrophic lake has a perimeter of 16.7 km, an area of 5.24 km$^2$, a volume of 166 million m$^3$ and a maximal depth of 71 m. During the year, the natural range of shoreline variation is about 2 m. This monomictic lake exhibited thermal stratification from April to November in 2019 [50].

Lake Serre-Ponçon is a hydroelectric dam reservoir, built on the Durance, a few metres after the Ubaye confluence, at an altitude of 779 m. This oligotrophic dimictic lake has a maximal volume of 1.27 billion m$^3$, a perimeter of 103.2 km, an area of 27.9 km$^2$ and a maximal depth of 129 m. It exhibits an annual water-level variation of about 23 m and thermal stratification from April to October.

Lake Etang des Aulnes is a natural, shallow freshwater lake surrounded by wetlands, at an altitude of 11 m. This eutrophic lake has a perimeter of 5 km, an area of 1 km$^2$, a volume of

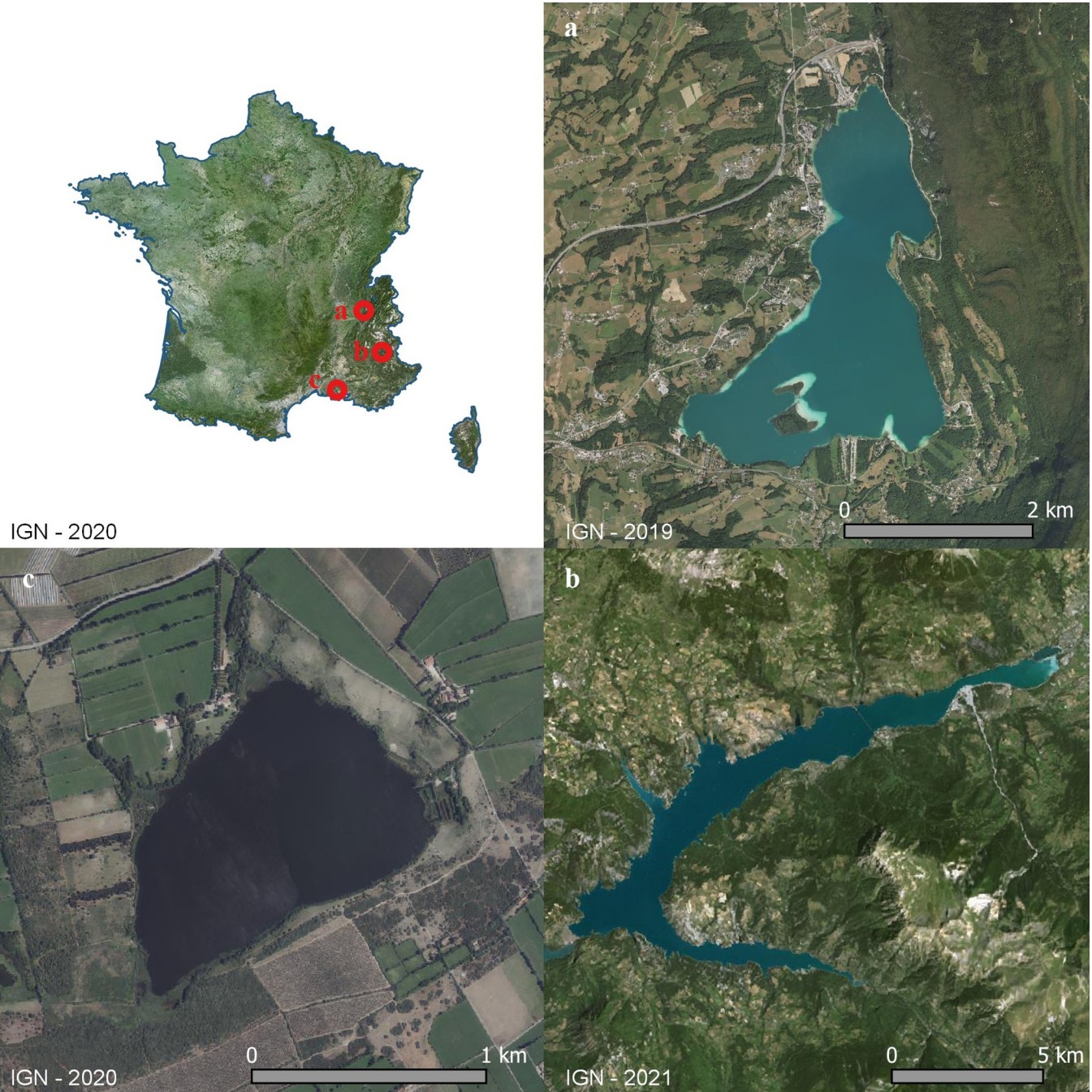

**Fig 1.** Locations of the lakes: (a) Aiguebelette, (b) Serre-Ponçon, (c) and Etang des Aulnes (IGN, 2019–2021).

3.34 million m$^3$ and a maximal depth of 5.5 m. The annual variation of the shoreline is about 0.6 m, without stratification over the year.

Field site access was approved by the water managers of each lake: Communauté de Communes du Lac d'Aiguebelette (CCLA) for lake Aiguebelette, Syndicat Mixte d'Aménagement et de Développement de Serre-Ponçon (SMADESEP) and by Conseil Départemental des Bouches-du-Rhône for lake Etang des Aulnes. No specific authorization were necessary to filter water.

## Gillnet fishing data

We used data obtained from national fishing surveys conducted by the French Biodiversity Agency (OFB) and water agencies as part of the WFD monitoring programme. Fish data were collected according to CEN standard [51]. This protocol defines the number of benthic and pelagic gillnets (composed of a standardised number of panels of different mesh sizes) to use depending on the lake area and the maximum depth. Between July and October, the gillnets were placed in the evening and remained in place one night before being lifted following the sampling recommendations to cover the maxima of fish activity. Fish were identified to the species level. For each lake, we gathered data from two fishing campaigns with at least 4 years between campaigns (S1 Table). We considered all the species caught during these campaigns to define the species composition detected by traditional methods.

## eDNA sampling protocol

For each site, eight sampling campaigns were performed throughout 1 year with a time lag of 6 weeks between two campaigns. Due to logistic constraints, the beginning and the end of the campaigns varied between lakes: 9 April 2019 to 27 February 2020 for Lake Etang des Aulnes, 10 April 2019 to 25 February 2020 for Lake Serre-Ponçon and from 23 May 2019 to 10 March 2020 for Lake Aiguebelette. For the last lake, only seven campaigns were performed; the last campaign was cancelled because of the COVID-19 pandemic.

The integrative sampling strategy was performed in continuous mode along a long transect (from hundreds of metres up to 2.5 km) by filtering a large volume of water for 30 min or until clogging (from 17.5 L to 47.5 L). This integrative sampling allowed us to sample a large variety of habitats and to reduce the species detection biases due to local variation [52]. For each lake and each campaign, we realised three surface transects by boat: one on the left shore, one on the right shore and one above the deepest area (hereafter called the 'centre transect'). As far as possible, between two sampling sessions, the location of each transect was kept constant. Transects did not overlap each other and were chosen to not cross tributaries. Water was pumped continuously from 10 to 15 cm above the water surface using a peristaltic pump (Vampire sampler, Bürkle, Germany) and then filtered (tangential filtration) using a VigiDNA 0.45-μm filter capsule (SPYGEN, France) [33]. Water was taken directly from the lake to the cartridge through a strainer and a plastic tub that were both single-use and sterile.

One sample was taken in the deepest zones of each lake (5–10 m from the bottom for the deepest lakes, 1 m for Lake Etang des Aulnes) using a Niskin water sampler of 5 L. Six point samples were collected in spatially close points and pooled to obtain a 30-L sample filtered with the VigiDNA capsule. Between two sampling campaigns, the Niskin was disinfected in a bath of bleach (0.5%) for 24 h.

At the end of each filtration, the capsule was emptied of water and filled with 80 mL of CL1 Conservation buffer (SPYGEN), shaken for 1 min and stored at room temperature.

## Extraction, amplification protocol, high-throughput sequencing and bioinformatics analysis

DNA extraction, amplification using 'teleo' primers [29], high-throughput sequencing and bioinformatics analysis were performed following the protocol described by Pont et al. (2018) [40].

For DNA extraction, each filtration capsule was agitated for 15 min on an S50 shaker (CAT Ingenieurbüro™) at 800 rpm and the buffer was then emptied into a 50-mL tube before being centrifuged for 15 min at 15,000 × $g$. The supernatant was removed with a sterile pipette,

leaving 15 mL of liquid at the bottom of the tube, after which 33 mL of ethanol and 1.5 mL of 3M sodium acetate were added. The tubes were stored for at least one night at −20°C, centrifuged at 15,000 × *g* for 15 min at 6°C, and the supernatants were discarded. After this step, 720 μL of ATL buffer from the DNeasy Blood & Tissue Extraction Kit (Qiagen) was added. The tubes were vortexed, and the supernatants were transferred to 2-mL tubes containing 20 μL of proteinase K. They were finally incubated at 56°C for 2 h. Subsequently, DNA extraction was performed using NucleoSpin® Soil (Macherey-Nagel GmbH & Co., Düren, Germany) starting from step 6 and following the manufacturer's instructions. The elution was performed by adding 100 μL of SE buffer twice. After the DNA extraction, the samples were tested for inhibition by quantitative polymerase chain reaction (qPCR; Biggs et al. 2015). If the sample was considered inhibited, it was diluted fivefold before the amplification.

DNA amplifications were performed in a final volume of 25 μL, using 3 μL of DNA extract as the template. The amplification mixture contained 1 U of AmpliTaq Gold DNA Polymerase (Applied Biosystems, Foster City, CA), 10 mM Tris-HCl, 50 mM KCl, 2.5 mM MgCl2, 0.2 mM each dNTP, 0.2 μM "teleo" primers (Valentini et al., 2016), 4 μM human blocking primer for the teleo primers (Civade et al., 2016) and 0.2 μg/μL bovine serum albumin (BSA, Roche Diagnostic, Basel, Switzerland). The teleo primers were 5'-labelled with an eight-nucleotide tag unique to each PCR replicate (with at least three differences between any pair of tags), allowing for the assignment of each sequence to the corresponding sample during sequence analysis. The tags for the forward and reverse primers were identical for each PCR replicate. In total, 12 replicate PCRs were run per filtration.

Library preparation and sequencing were performed at Fasteris (Geneva, Switzerland). Four libraries were prepared using the MetaFast protocol (Fasteris, https://www.fasteris.com/dna/?q=content/metafast-protocol-ampliconmetagenomic-analysis), a ligation-based method, and then sequenced on four separated runs on a MiSeq (2 x 125-bp) (Illumina, San Diego, CA, USA) with the MiSeq Flow Cell Kit Version 3(Illumina). Eight negative extraction controls and two negative PCR controls (ultrapure water, 12 replicates) were amplified and sequenced in parallel to monitor possible contaminants.

Sequence reads were analysed using programmes implemented in the OBITools package (http://metabarcoding.org/obitools) [53] following a protocol described elsewhere [29]. Shortly, the forward and reverse reads were assembled using the *illuminapairedend* programme using a minimum score of 40 and retrieving only the joined sequence. The reads were then assigned to each sample using the *ngsfilter* programme. A separate data set was created for each sample by splitting the original data set into several files using *obisplit*. After this step, each sample was analysed individually before merging the taxon list for the final ecological analysis. Strictly identical sequences were clustered together using *obiuniq*. We discarded sequences shorter than 20 bp, or with an occurrence lower than 10, and labelled 'internal' with the *obiclean* programme that correspond most likely to PCR substitutions and indel errors. Taxonomic assignment of the molecular taxonomic units (MOTU) was performed using the programme *ecotag* with the local reference database Teleostei [29] and the sequences were extracted from the ENA Release 142 (standard sequences) database using the *ecopcr* programme [54]. MOTUs showing less than 98% similarity to the local reference database were removed. Finally, considering the bad assignments of a few sequences to the wrong sample due to tag-jumps [55], all sequences with a frequency of occurrence below 0.001 per taxon and per library were discarded. A supplementary filter was applied during data treatment to exclude species detected with less than two positive PCR replicates out of 48 per sampling site.

By using teleo primers, the following species are not differentiated [29] and are referred to as the genera: *Salvelinus* sp., *Cottus* sp. and *Gobio* sp. This is also the case for the allochthone

acclimated species of *Carassius* genus. The teleo barcode does not discriminate some species belonging to different genera. They are grouped into 'complexes' [26] (see S1 Text).

### Thermal stratification

To assess whether lakes were stratified, vertical temperature profiles were analysed for each sampling date. For Serre Ponçon and Etang des Aulnes, a multiparameter probe (ExO-2®, AnHydre) was used to estimate the vertical profiles. For Aiguebelette, vertical profiles were estimated using a multiparameter probe (SST–CTM214) by the Alpine Lakes Observatory (OLA; https://si-ola.inrae.fr/) [56].

### Statistical analysis

In the statistical analysis we considered both the species detections (presence 1, absence 0) and the number of DNA reads assigned to each species. The number of species detected in a sample was compared with the species richness obtained from gillnets. To avoid richness overestimation with eDNA, species complexes were discarded from the species richness computation when at least one species of the complex was detected individually. If some species were existent but not individually detected, this could have led to an underestimation of the richness.

To be comparable, the number of reads were standardised (Pont et al., 2018), by dividing them by the total number of reads observed in the sample. They were then rescaled to 100,000.

To assess whether contrasting fish communities (from the three lakes) could be distinguished by eDNA, we performed two non-metric multidimensional scaling (NMDS) analyses, one with the presence/absence data (Bray–Curtis distance matrix) and one with the standardised reads (Gower distance matrix) [57]. NMDS seeks to synthetise the information held in the distance matrix, by representing objects (eDNA samples) on a simplified graphical display that reflects at best (assessed through the stress statistic) the closeness between objects. The 'lake' effect was tested using permutational multivariate analysis of variance (PERMANOVA) [58].

For each lake, NMDS was performed on all the samples collected to assess the spatio-temporal dynamics of the eDNA signal. To visualise the temporal evolution between each campaign for each sampling location (the two shores, the centre and the bottom), the NMDS coordinate points were connected following the chronological order of the campaign.

The R software (4.1.1; [59]) was used for all analyses and graphics, with the packages vegan (2.5–7.; [60]) for the NMDS and PERMANOVA (anosim function) analyses and ggplot2 (3.3.3; [61]) for the graphics.

## Results

### Stratification

The thermal profiles of the lakes showed different dynamics. Aiguebelette and Serre-Ponçon were stratified, respectively, from April to November [50] and April to October. Temperatures along the vertical profile of Etang des Aulnes were homogeneous during all of the campaigns, and the water appeared to be mixed regularly (S1 Fig).

### Gillnets captures

The number of species caught with gillnets was, respectively, 14, 14 and 12 at Aiguebelette, Serre-Ponçon and Etang des Aulnes. These numbers represent the total number of species observed when considering all the sampling campaigns available. They were not reached with only one campaign. Despite a common pool of species caught between campaigns, a variability in the species captured was noticed (S1 Table).

In Aiguebelette, the number of species captured was 12 in 2009, 13 in 2014 and 12 in 2020. In 2014, a unique individual of *Cyprinus carpio* (L., 1758) was collected and not found again in 2020.

In Serre-Ponçon, only 12 species were captured in 2011 and 11 in 2017. One individual of *Salvelinus umbla* (L., 1758) and two of *Scardinius erythrophthalmus* were found in 2014, but these species were not caught in 2017. On the other hand, one individual of *Blicca bjoerkna* and one individual of *Tinca tinca* (L. 1758) increased the inventory of the 2017 campaign.

In Etang des Aulnes, 11 species were captured in 2011, including *Carassius carassius* and *Abramis brama*. Surprisingly, although 49 individuals of *A. brama* were captured in 2011, none was found in 2015, excluding common bream from the taxonomic inventory. *C. carpio* was found only in 2015 with one individual.

## Taxa recorded

The sequencing yielded a total of 48,974,572 raw reads, with 11,986,001 raw reads for fish taxa in Lake Aiguebelette, 16,584,986 in Lake Serre-Ponçon and 20,403,585 in Etang des Aulnes before bioinformatic filtering.

Finally, 30 different freshwater fish taxa (species, genera and complexes) were detected, representing 27,286,801 raw reads after bioinformatic filtering, which corresponds to 9,200,000 standardised reads. The European bass *Dicentrarchus labrax* (L. 1758) marine species was detected in Etang des Aulnes and removed from the data, since this detection was probably linked to human consumption.

The greatest number of taxa detected was for Aiguebelette ($N = 19$), followed by Serre-Ponçon ($N = 18$) and Etang des Aulnes ($N = 17$). These detections surpass the number of species collected with WFD standardised gillnets, with 14, 14 and 12 species caught, respectively. Almost all species collected with gillnets were detected by eDNA. Only *B. bjoerkna* was not detected with eDNA in Serre-Ponçon and in Etang des Aulnes, while the morphologically close species *A. brama* was detected by eDNA but not recorded in the gillnet data (Fig 2). Some species were only detected by eDNA: the benthic species *Salaria fluviatilis* (Asso, 1801) and *Barbatula barbatula* (L., 1758), the small, invasive introduced species *Pseudorasbora parva* (Temminck & Schlegel, 1846), the anadromous *Anguilla anguilla* (L., 1758) and the pelagic species *Sander lucioperca* (L., 1758). Otherwise, all common lacustrine species were detected by the two methods, for example pike *Esox lucius* (L., 1758) or common perch *Perca fluviatilis* (L., 1758).

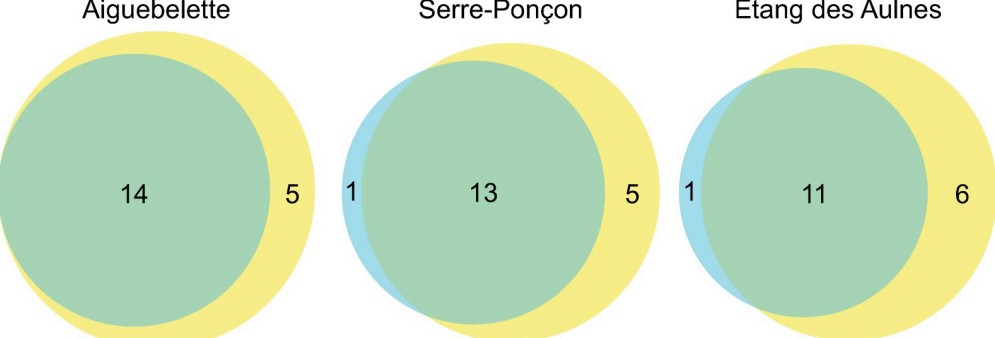

**Fig 2.** Venn diagram showing number of species detected by eDNA, all campaigns together (yellow) and found with WFD standardised gillnets (two campaigns per lake) (blue) at the three study sites: (A) Aiguebelette, (B) Serre-Ponçon, (C) Etang des Aulnes.

Although globally more species were detected with eDNA than gillnets, this was not true for all locations nor for all campaigns (Fig 3). In Aiguebelette, from May to September and in March, with all locations combined (the two shores, the centre and the bottom), the number of detected species with eDNA was higher or equal to that found with gillnets. If the bottom samples are not taken into account, this pattern changes somewhat, with only four campaigns out of seven in which a higher species number was detected with eDNA than with gillnets. The maximum detection in surface was obtained during late spring, summer and late winter in this lake. For Serre-Ponçon, the results from the gillnet method were never surpassed with all sampling locations combined. There was a great decrease in the number of species detected between summer and autumn for almost all locations, as well as in winter. Finally, in Etang des Aulnes, when combining surface samples, the number of species detected with eDNA was higher or equal to that detected with gillnets for all campaigns from April to August. Indeed, the number of species detected decreased in autumn and winter, with a peak during the November campaign.

This pattern showed a variability over time and space for the three sites, which we analysed in more detail for each lake.

## Comparison of sites

Because the patterns observed in the species detection and in the standardised reads were very similar, only the results for the species detection (presence–absence) are displayed (but see S2 Fig for the reads).

The NMDS based on all eDNA samples (Bray–Curtis distance, stress = 0.156) showed that the fish fauna of the three sites were significantly different whatever the sampling location and period (PERMANOVA, $p < 0.001$; Fig 4) as revealed by the non-overlapping point clouds. Due to these differences, the spatio-temporal dynamics of the eDNA signal was studied for each site independently.

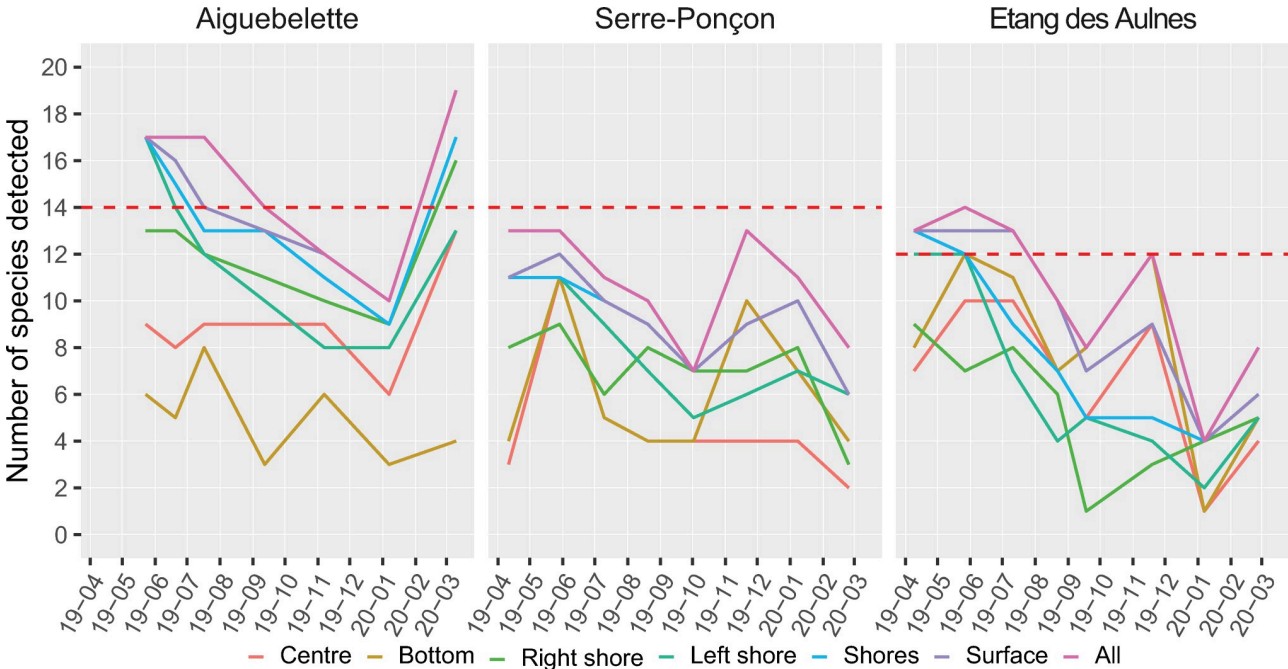

**Fig 3.** Number of species detected in (A) Aiguebelette, (B) Serre-Ponçon and (C) Etang des Aulnes, during each campaign (year–month), for each location. For comparison, the number of species found during the two latest gillnets campaigns is given in dashed red.

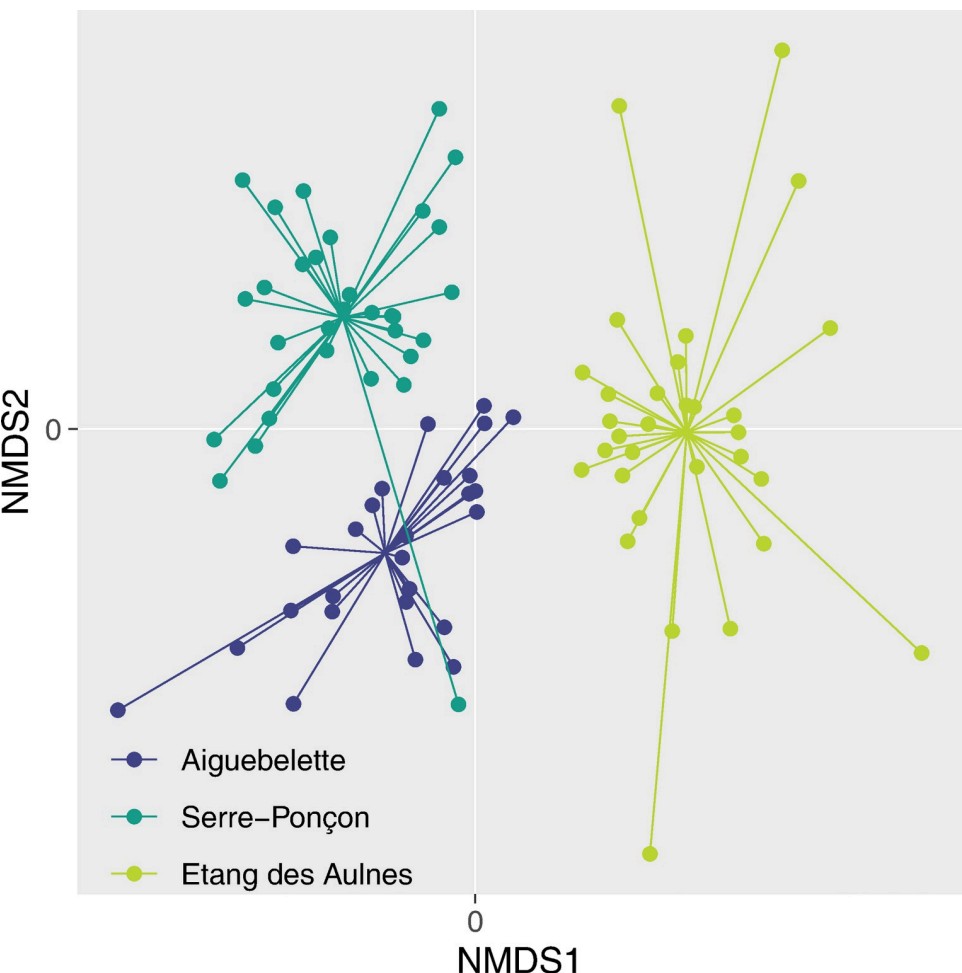

**Fig 4. NMDS ordination of fish assemblages of all eDNA samples from Aiguebelette, Serre-Ponçon and Etang des Aulnes in each site.** Each sample was connected to the central position of the lake where it was collected (average locations).

### Spatio-temporal dynamics

In Aiguebelette, when analysing the samples of all seven campaigns, the NMDS (Bray–Curtis distance, stress = 0.132) showed a clear difference of fish assemblage for the three locations: bottom, surface centre, and shores (right and left together) (PERMANOVA, $p < 0.01$; Fig 5A). The bottom samples were the only ones in which *Salvelinus* sp. (except in November) and European whitefish (*Coregonus lavaretus* (L., 1758)) were always detected. The surface samples were very different from the bottom ones, and, although a spatial structure was clear between the centre and the shores, they both showed a very particular assemblage with the outsider point in January 2020. Compared to the shore assemblages, the samples from the centre yielded a different signal with specific species detected during the campaigns, for example *C. lavaretus* in June 2019 or *E. lucius* in January 2020. From May to November, the list of species detected in the centre was relatively stable, as revealed by the close locations on the NMDS scale of these samples (Fig 5A). Samples from the shores exhibited relatively similar temporal trajectories, especially the winter assemblages (January and March) (Fig 5A).

In Lake Serre-Ponçon, the NMDS based on all samples (Bray–Curtis distance, stress = 0.180) showed that the fish assemblages of the four locations were significantly different

(PERMANOVA, *p* = 0.002; Fig 5B), even though the point clouds are overlapping. As seen in Aiguebelette, on the shores in spring and in summer the samplings were close to each other, reflecting similar assemblages over time in those locations. Singularities appeared from November to the last campaign in February for the left shore, and solely in February for the right shore. The difference with Aiguebelette is evident for the centre sample, which varied greatly during the survey and did not maintain a specific assemblage through time. Regarding the fauna differences between the locations, the centre sampling yielded the detection of *C. lavaretus* during the two spring campaigns (April and May 2019) in comparison with the shore samplings. However, this species was found there during each campaign from November 2019 to February 2020. On the other hand, compared to the surface samplings, the bottom revealed very diverse species signals during the year. Some species are known to be present in the lake but are undetected elsewhere, such as *Silurus glanis* (L., 1758), some are affiliated to lotic environments, such as *Rhodeus amarus* (Bloch, 1782), and some are ecologically expected on shores but are not found there, for example *Tinca tinca*.

Contrary to the other sites, for the Etang des Aulnes, the NMDS (Bray–Curtis distance, stress = 0.157) showed no differences in the fish fauna between the four locations (PERMANOVA, *p* = 0.232; Fig 5C). It should be noted that during two campaigns (August 2020 and February 2021), several filtration capsules were clogged or almost clogged before the end of the 30 min, leading to lower filtered volumes. Here, the sampling results are extremely variable over time, regardless of the location. However, it should also be noted that winter samples were singular such as in the other lakes. Although centre sampling did not yield unique information concerning fish assemblages in comparison with the shore sampling, the bottom sampling provided a signal of a particular species, *Salvelinus* sp., only found here. The other species detected in the bottom samples at one time could be found in surface samples during another campaign.

## Spatio-temporal variabilities of species-by-species read counts

For the majority of species (all figures in S3 Fig), the dynamics of their respective standardised reads showed a temporal and spatial evolution that matched their known ecology. For example, for the European whitefish *C. lavaretus* (Fig 6.1), in Aiguebelette and Serre-Ponçon, a peak of detection occurred during winter on shores corresponding to the reproduction period and spawning area of this species.

The main difference between the two lakes is the dynamics of the number of standardised reads for the bottom samples. In Aiguebelette, the bottom of the lake was the location where European whitefish was mostly detected, throughout the year, and in Serre-Ponçon, the bottom samples showed a reduced frequency of detection for this species, with a peak during winter.

As expected, *P. fluviatilis*, a very common lacustrine species in France, was detected during 45all of the campaigns in the three lakes and in different locations (Fig 6.2). In Aiguebelette, in every location, the number of standard reads increased from spring to summer, and decreased in winter. In Serre-Ponçon, the detection fluctuated greatly along time and space, but, except for the bottom sampling in January 2020, perch was detected during every campaign in every sample. In Etang des Aulnes, except during spring, *P. fluviatilis* was never detected on the left shore. However, when all the surface samples (centre, left shore, right shore) were combined, this species was detected during every campaign, showing stability in detection among the surface samples.

## Discussion

The results of the present study not only confirmed the power of eDNA metabarcoding for assessing the species composition of lacustrine fish assemblages, but they also illustrated how

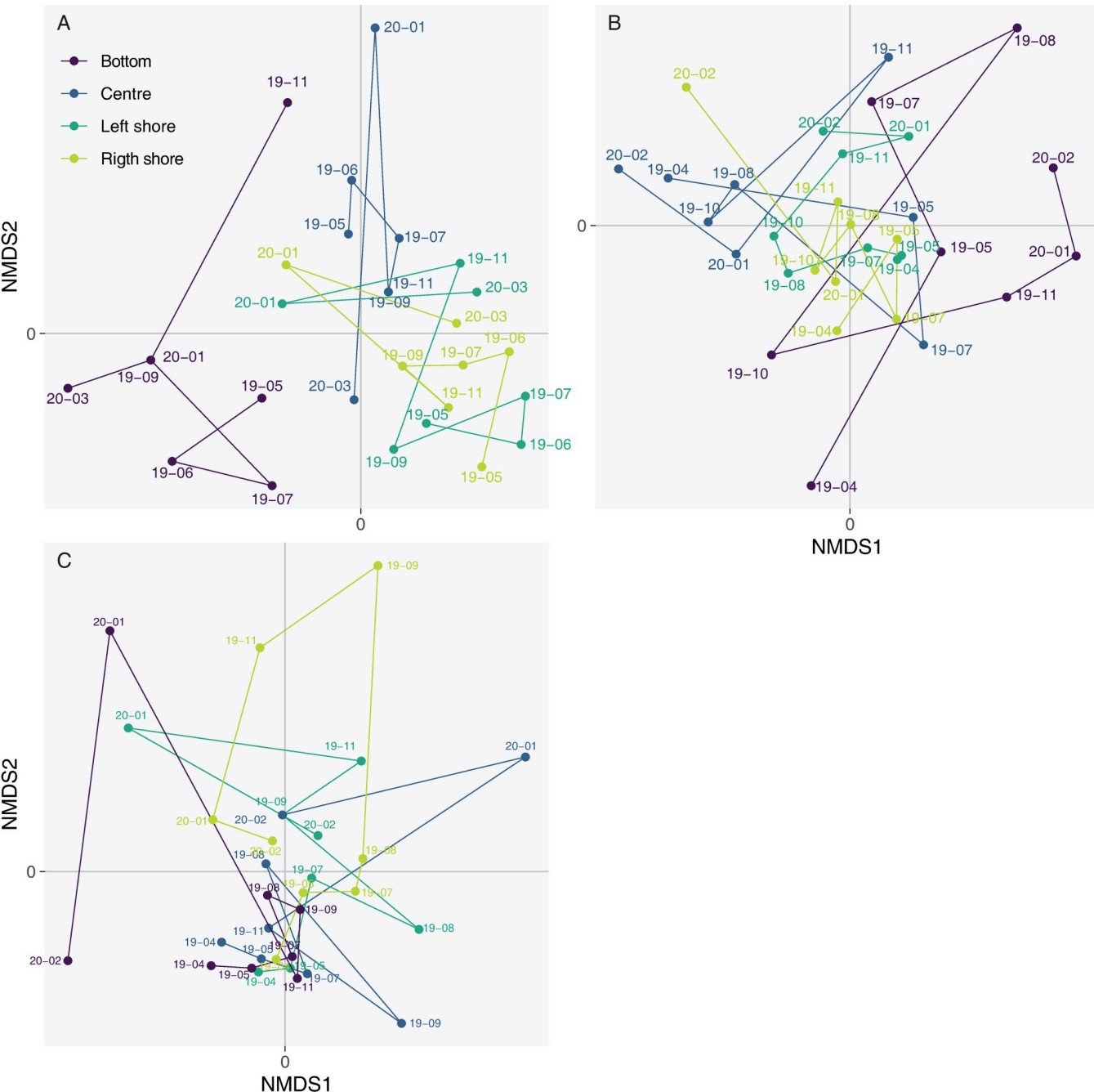

**Fig 5.** NMDS ordination of fish assemblages for each eDNA sample (dots) in (A) Aiguebelette, (B) Serre-Ponçon and (C) Etang des Aulnes. For each sampling location (left shoreline, right shoreline, central location and depth), dots are connected to represent a temporal pathway (from the first to the latest sampling date).

the fish eDNA signal varied over time and space according to the physical characteristics of the lakes (from shallow and small to deep and large lakes) and the fish fauna.

For the three lakes studied here, we confirmed the capacity of eDNA to distinguish contrasting fish communities, as already reported for riverine and lacustrine communities in several studies [40, 47, 62]. The value of this method in discriminating between communities was

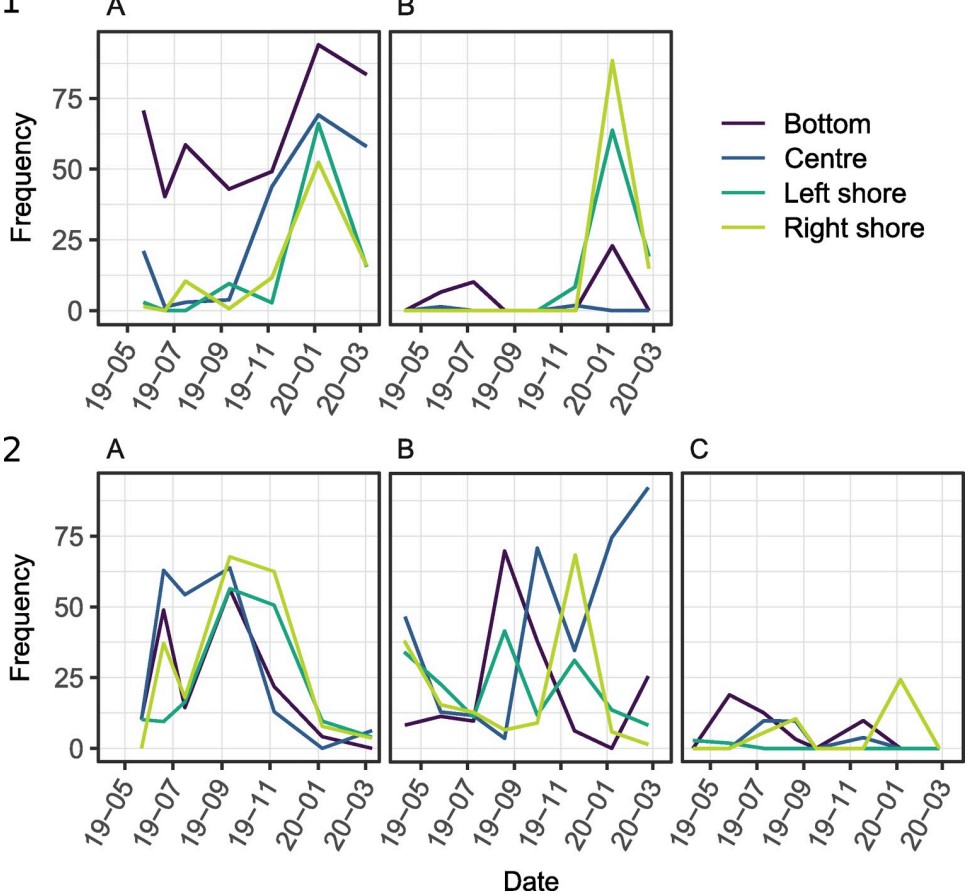

**Fig 6.** Frequency of reads for (1) *Coregonus lavaretus* and (2) *Perca fluviatilis* during each campaign in (A) Aiguebelette, (B) Serre-Ponçon and (C) Etang des Aulnes, in the four locations.

confirmed here for all sampling locations and times (sampling date). Lakes Aiguebelette, Serre-Ponçon and Etang des Aulnes were selected because they display diverse environmental conditions and lake functioning. Surveying simultaneously the different types of lake using this spatio-temporal sampling effort constituted a step further into the understanding of the capacity of eDNA to characterise the fish assemblages from only one eDNA sampling campaign.

When considering all the eDNA samples collected, for any of the three lakes, we observed that eDNA allowed us to detect all the species that were also detected with the traditional methods (gillnets), in accordance with previous studies [28, 38, 63]. However, when considering each sampling campaign independently, we found that the number of species detected by eDNA could be lower than the number of species detected with gillnets (i.e. species observed in the two latest WFD surveys). This pattern was mainly observed during autumn and early winter campaigns, when the lowest numbers of species were obtained, whatever the lake or location sampled.

The only species detected solely with gillnets was *B. bjoerkna*, but with a taxonomic assignment to the complex '*A. brama/B. bjoerkna*' with eDNA. The difficulty to assign, with the teleo barcode, a species name to these two species was already known [40]. Nonetheless, even if some species are only detected in complexes, eDNA enabled the detection of species not

caught by gillnets, as already mentioned for different water bodies [33, 35]. Such additional detections always raised the suspicion of possible false-positives (species detected but not occurring in a site), a limitation of molecular methods [64]. In our survey, we know that the species found with eDNA were actually present because they were detected with other traditional methods (i.e. electrofishing and/or fyke nets; Westrelin unpublished data). In Etang des Aulnes, this was the case for the small species *Gambusia holbrooki* (Girard, 1859) and *Pseudorasbora parva*, which is already very invasive in France [65]. The benthic *Salaria fluviatilis*, detected in Aiguebelette and Serre-Ponçon (Salmon, personal communication), was never observed in gillnets while it was very well detected with eDNA metabarcoding. Its low capturability by gillnets is likely related to its small size and habitat [7] as this species lives under rocks on the shorelines. eDNA methods seem to overcome some of the limitations of the gillnet method, including the detection capacities for species with particular ecology or behaviour that generally prevents their capture with traditional methods. eDNA is also an excellent tool with which to detect rare species that cannot be found otherwise [28, 63], as we observed here for *Silurus glanis* in Aiguebelette and Serre-Ponçon. In this reservoir, water managers were sceptical about the presence of catfish when it was first detected with eDNA (Salmon unpublished data), but an individual stranding on the shore confirmed the capacity of early detection with eDNA [26, 63, 66].

In terms of the spatial distribution of the eDNA signal, a clear difference was observed between the three sites. In the small and shallow Etang des Aulnes, no clear spatial organisation was noticed, although the greatest number of species was detected in samples from the deep zones, for all campaigns combined. All of the species found in these samples were also found in the other locations (sub-surface sampling of shores and lake centre) at different times of the year, except for *Alburnus alburnus* and *Salvelinus* sp. Concerning the latter, its presence was very unlikely in this lake where temperatures are too high for this species which is mostly found in cold lakes [67]. Since this species was found upstream in the watershed [40, 68], the eDNA detection in the lake can be an exogenous input, probably brought by affluent or by avian faeces [47, 69]. Sediment resuspension could also explain this pattern because eDNA is potentially well conserved inside [13]. Deep-water samplings were conducted for the three study lakes that exhibited contrasting morphometry. Our results suggest that deep-water sampling is not pertinent nor necessary for small and shallow waterbodies such as Etang des Aulnes. Since water mixing in lakes helps to homogenise the eDNA signal along the water column [46, 52], deep-water samples are not useful for shallow lakes which are regularly mixed. It is possible, even recommended, to adapt the sampling method to the type of lake when establishing a standardised protocol. For instance, the CEN protocol does not require pelagic gillnets for lakes shallower than 20 m. For the deepest monomictic lakes, bottom samples were less speciose than surface samples, with almost all of the species detected in deep samples also detected in sub-surface samples (on shores and/or centre), in accordance with previous observations from English lakes [39]. When these monomictic lakes are stratified, the fish eDNA is not distributed homogeneously among the water column and it follows the vertical distribution of species [39, 45, 46]. When waters are mixed, the signal has, on the contrary, a homogeneous vertical distribution. Consequently, the relevance of collecting deep-water samples for biodiversity monitoring is also questionable for deep lakes, and further investigations on a larger diversity of lakes, such as deep monomictic lakes, are necessary.

eDNA signal also has a clear spatial variability when considering surface samples [39, 70]. Shorelines in particular appeared to be the locations with the greatest detectable species richness, as seen here in Lake Aiguebelette and elsewhere [39, 45]. Shores are shallow zones hosting a diversity of habitats, known to play a major role in species distribution [41, 71] and thus in eDNA distribution [47, 72]. This variability of habitats has an impact on diverse life

components of fish such as nourishment [73, 74], rest [75], protection from predators [75] or reproduction [76, 77]. It leads to a differential use of the space over time, depending of the needs and constraints faced by individuals. In Lake Aiguebelette, despite a common pool of species, the eDNA signal was not distributed homogeneously between the two shores. The left shore is dominated by vegetation and rocks, and species such as pike (*Esox lucius*) [78] or freshwater blenny (*Salaria fluviatilis*) [79] were detected. This is consistent with their use of such habitats to spawn or to protect larvae. On the right shore, *Lepomis gibbosus* (L., 1758) was detected, as it offered shallow sandy or gravel habitats used by this species to nest its eggs [80]. The few differences between shores illustrated the differences in the present species and their use of habitats, which support the good match between local habitat and eDNA signal detection [62]. In the relatively homogeneous sampling area of Serre-Ponçon, the shoreline displayed poor fish assemblages.

Because of their position in the river network and their functioning, reservoirs are different from natural lakes in terms of eDNA signal distribution. In Serre-Ponçon, the portion of shore that was sampled hosted very homogeneous rocky habitats due to the former valley slopes and to water-level fluctuations. Frequent water-level fluctuations hinder vegetation settlement on shores [81–83] and homogenise habitats. This limits the sustainable establishment of a diverse community on shores, thereby affecting eDNA distribution. Thus the spatial distribution of eDNA in reservoirs is directly related to their hydrodynamics due to the constraints on the shores but also to the hydrology of the upstream contributors [62] (e.g. Durance and Ubaye). The lotic species, *Cottus* sp., which is not known to occur in the lake but in its affluents (Durance, Ubaye), was detected in samples from deep zones but never on the shores. This suggests that DNA from riverine species could be transported from the tributaries [40, 84]. During the stratification of the lake, the difference in temperatures between the lake and its affluent could lead riverine waters, loaded with sediments and suspended particles [85], to go into the hypolimnion. Such water movements are influenced by the hydrological regime of the river, e.g. during the ice melting period for Durance [86]. The hydrodynamics of reservoirs is important when considering the spatial and vertical distribution of the fish eDNA signal [87] because of water movements and habitat availability over time.

Abiotic variability is thus a key factor for understanding the temporal dynamics of the fish eDNA signal within lakes. However, despite the substantial environmental differences between the study sites, a common peak of detection occurred in spring and late summer. This peak matches the reproduction period of most of the lacustrine species detected, suggesting that species phenology plays a major role in the temporal variability of eDNA signal. eDNA concentration followed a strong seasonal variation, with a peak of eDNA concentration or number of reads observed during fish reproduction [88, 89]. This was also observed with metabarcoding data for late autumn/winter spawners, such as *Coregonus lavaretus*. Similarly to what was observed for the arctic char (*Salvelinus alpinus*) by Di Muri et al. (2020) [90], we did not detect whitefish on the shorelines outside their period of reproduction, but a peak of DNA reads was found in January corresponding to their maximum reproductive activity. This species moves from pelagic areas to the shores during winter to find suitable spawning habitats, explaining such riverine detections.

The difference in species detectability over time, due to the phenology and ecology of the species, raises the question on the most appropriate temporal window for sampling eDNA to characterise lacustrine fish diversity. This is a common question when developing or standardising new protocols. Sampling in spring on shorelines would lead to better detection and thus a more consistent assessment of fish communities dominated by spring spawners (most of the French lakes). However, this would imply a lower detectability of salmonids and coregonids and thus a possible bias in biodiversity assessment for cold lakes (mostly high-elevation lakes

in France). Elsewhere, sampling in winter would lead to a lower estimation of fish biodiversity for most French lakes surveyed for the WFD (lakes larger than 50 ha). Due to the significant diversity of fish assemblages in French lakes, it is difficult to determine a unique temporal window for all. Adapting the temporal window is a possibility that could be considered but this must be done on the basis of environmental criteria that have to be defined (e.g. temperature threshold). These criteria could include the warmest periods, during microorganism blooms, or after meteorological events that could increase the amount of particles in suspension in the water, to avoid the clogging of filters as we experienced in our study during August or February in Etang des Aulnes. However, using biological criteria would require prior knowledge on fish communities.

The sampling period could also change according to the objectives of the surveys. Using eDNA biomonitoring results, the aim could be the assessment of population health or the ecological status of waterbodies [40, 91]. Such assessment generally involves the estimation of fish abundance [4]. Several studies investigated eDNA quantification, searching for relationships between eDNA signal found with metabarcoding (concentration, number of reads) and biomass or abundance [92, 93]. However, this estimation based on the number of sequences can be biased by the presence of semen [94] or juveniles hatching [95], which is known to increase the amount of DNA released in the environment during reproduction and the larvae growing season. Despite a high recruitment, a large number of juveniles may not be able to reach the first year, which means there are not as many individuals at the end of the growing season. For population monitoring, it seems better to wait until the ontogenetic shift and mortality phenomenon of the year cohort, as advocated in the CEN protocol. In this case, as seen previously, the mixing period for stratified lakes was an interesting time to sample, occurring after the ontogenetic shift for most of the lakes and leading to a homogeneous repartition of eDNA in the lake.

## Supporting information

**S1 Table. CEN gillnets campaigns for the lakes Aiguebelette, Serre-Ponçon and Etang des Aulnes.**
(DOCX)

**S1 Text. List of the complexes of species detected by teleo barcode.**
(DOCX)

**S1 Fig. Vertical profiles of lake Serre-Ponçon, lake Etang des Aulnes and lake Aiguebelette.**
(PDF)

**S2 Fig. NMDS based on the standardised reads data.**
(PDF)

**S3 Fig. Spatio-temporal variability of the eDNA signal for each species.**
(PDF)

**S1 Appendix. IGN answer concerning the compatibility of the licence Etalab 2 with the CC BY licence.**
(PDF)

## Acknowledgments

We would like to thank the local politicians and organisations of Lake Serre-Ponçon (SMADE-SEP "Syndicat Mixte d'Aménagement du Lac de Serre-Ponçon", Fishing Federation of

Hautes-Alpes and Alpes-de-Haute-Provence, Hautes-Alpes service of the OFB SD05), of Lake Aiguebelette (CCLA "Communauté de communes du Lac d'Aiguebelette") and of Etang des Aulnes (Bouches-du-Rhône French department). We would also like to thank for the technical support during field sampling René Conraud from OFB SD05 and Dewis Davudian from the Fishing Federation of Serre-Ponçon, Pascal Perney and Jean-Christophe Hustache from INRAE UMR CARRTEL on Aiguebelette, and Ange Molina, Julien Dublon and Julien Bocchino from INRAE RECOVER on Etang des Aulnes. We also thank SPYGEN staff for technical support during eDNA analysis. We are further grateful to Julien Dublon for his advice concerning location choices on Serre-Ponçon, and to Quentin Salmon and Samuel Westrelin for the personal communication and for their expertise. We are grateful to Laurent Valette from INRAE RiverLy for his precious help concerning GIS data and their licence. We also thank Isabella Athanassiou for English correction.

## Author Contributions

**Conceptualization:** Isabelle Domaizon, Jean-Marc Baudoin, Tony Dejean, Maxime Logez.

**Formal analysis:** Alix Hervé, Alice Valentini, Maxime Logez.

**Funding acquisition:** Jean-Marc Baudoin, Tony Dejean.

**Investigation:** Alix Hervé, Pierre Gibert, Pauline Jean, Tiphaine Peroux, Jean-Claude Raymond, Marine Vautier, Maxime Logez.

**Methodology:** Jean-Marc Baudoin, Tony Dejean, Pauline Jean, Tiphaine Peroux, Alice Valentini.

**Project administration:** Jean-Marc Baudoin, Tony Dejean, Maxime Logez.

**Resources:** Pierre Gibert, Pauline Jean, Tiphaine Peroux.

**Software:** Alice Valentini.

**Supervision:** Isabelle Domaizon, Jean-Marc Baudoin, Tony Dejean, Maxime Logez.

**Validation:** Jean-Claude Raymond, Alice Valentini.

**Visualization:** Maxime Logez.

**Writing – original draft:** Alix Hervé, Alice Valentini, Maxime Logez.

**Writing – review & editing:** Isabelle Domaizon, Jean-Marc Baudoin, Tony Dejean, Alice Valentini.

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
