## [Decision Letter · Decision Letter 0]

15 Jun 2022

PONE-D-22-05624Spatio-temporal variability of eDNA signal and its implication for fish monitoring in lakesPLOS ONE

Dear Dr. Logez,

Thank you for submitting your manuscript to PLOS ONE. After careful consideration, we feel that it has merit but does not fully meet PLOS ONE’s publication criteria as it currently stands. Therefore, we invite you to submit a revised version of the manuscript that addresses the points raised during the review process.

We look forward to receiving your revised manuscript.

Kind regards,

Hideyuki Doi

Academic Editor

PLOS ONE

Journal Requirements:

2. We note that Figure (1) in your submission contain copyrighted images. All PLOS content is published under the Creative Commons Attribution License (CC BY 4.0), which means that the manuscript, images, and Supporting Information files will be freely available online, and any third party is permitted to access, download, copy, distribute, and use these materials in any way, even commercially, with proper attribution. For more information, see our copyright guidelines: http://journals.plos.org/plosone/s/licenses-and-copyright.

1. You may seek permission from the original copyright holder of Figure (1) to publish the content specifically under the CC BY 4.0 license. 

3. PLOS journals require authors to make all data underlying the findings described in their manuscript fully available without restriction unless the data are subject to ethical restrictions or owned by someone other than the authors (https://journals.plos.org/plosone/s/data-availability#loc-acceptable-data-access-restrictions). Therefore, we ask that you please upload underlying data to an appropriate data repository and update your Data Availability Statement accordingly or provide all contact details for where an interested researcher would need to apply to gain access to the relevant data. Please note that it is not acceptable for an author to be the sole named individual responsible for ensuring data access.

4. In your Methods section, please provide additional information regarding the permits you obtained for the work. Please ensure you have included the full name of the authority that approved the field site access and, if no permits were required, a brief statement explaining why.

5. In your Methods section, please include a comment about the state of the animals following this research. Were they released, euthanized or housed for use in further research? If any animals were sacrificed by the authors, please include the method of euthanasia and describe any efforts that were undertaken to reduce animal suffering

6. We note that you have indicated that data from this study are available upon request. PLOS only allows data to be available upon request if there are legal or ethical restrictions on sharing data publicly. For more information on unacceptable data access restrictions, please see http://journals.plos.org/plosone/s/data-availability#loc-unacceptable-data-access-restrictions. 

Additional Editor Comments (if provided):

I got the recommendations and comments from an expert reviewer in the field. The reviewer agreed that the manuscript is technically sound and the data support the conclusions.

However, the lack of an explanation in the Methods and Results sections was suggested, and I share the comments. Therefore, I can invite you to submit a revised version of the manuscript that addresses the points raised by the reviewers.

Reviewers' comments:

Reviewer's Responses to Questions

**Comments to the Author**

1. Is the manuscript technically sound, and do the data support the conclusions?

Reviewer #1: Yes

2. Has the statistical analysis been performed appropriately and rigorously? 

Reviewer #1: Yes

3. Have the authors made all data underlying the findings in their manuscript fully available?

Reviewer #1: Yes

4. Is the manuscript presented in an intelligible fashion and written in standard English?

Reviewer #1: Yes

5. Review Comments to the Author

Reviewer #1: Comments:

In this study, the authors used eDNA and assessed the spatio-temporal variability of its signal for fish monitoring in lakes.

I suggest publishing the paper after some revisions.

L27: May authors change to "6 weeks" instead of "2 months" (according to Line 154)

L32: maybe authors present here the traditional method used for the comparison.

L77: May authors change fish to eukaryotic species

L84: change to "The number of species that can be detected"

L95: "focused" instead of "focussed"

L97: "both of the two aspects" instead of "the two apsects together"

L101-102: change to "considering possible changes over a year"

L127: The subfigures may be named as (a) for Aiguebelette, (b) for Serre-Ponçon, and (c) Etang des Aulnes, instead of 1, 2, 3 which is a bit confusing

L235-241: This paragraph is results and not material and methods

L276: The thermal profile of Aiguebelette lake is missing from the Supplementary material.

L279, 285, 340: Define the specific Supplementary Material (eg 1, 2..) in each case

L293-295: This paragraph is a bit confusing. Someone cannot understand immediately why authors refer to Carassius carassius and Abramis brama. The first one is an allocthonous species and the reference in the second species is explained in the next sentece, however someone should know that the common bream is Abramis brama. Please be more clear in this paragraph.

Line 344-345: A comment about what the study of each site reveals (i.e. Fig 4), is missing

2 more general comments:

1. In eDNA studies it is common to use technical repetitions for the samples. In this study, the authors filtered a big volume of water and further they evaluated all the results from all the sampling campaigns in each lake. Therefore, I understand why they chose this specific approach. However, authors may explain a bit more why they followed this sampling strategy.

2. It is proved in many eDNA studies that the temperature affects the eDNA. In this study it seems that it is not the case. However, I think that a discussion about this issue is missing.

6. PLOS authors have the option to publish the peer review history of their article (what does this mean?). If published, this will include your full peer review and any attached files.

Reviewer #1: No

---

## [Author Response · Author response to Decision Letter 0]

14 Jul 2022

Dear Dr. Logez,

Thank you for submitting your manuscript to PLOS ONE. After careful consideration, we feel that it has merit but does not fully meet PLOS ONE’s publication criteria as it currently stands. Therefore, we invite you to submit a revised version of the manuscript that addresses the points raised during the review process.

We look forward to receiving your revised manuscript.

Kind regards,

Hideyuki Doi

Academic Editor

PLOS ONE

Journal Requirements:

We have modified the manuscript to meet PLOS ONE’s requirements. We have thus modified the style of the headings, the manner to refer to figures, tables, and to the supplementary material file. Supplementary Material captions and numbering were also changed. Reference numbers were placed in brackets (and not in parentheses).

2. We note that Figure (1) in your submission contain copyrighted images. All PLOS content is published under the Creative Commons Attribution License (CC BY 4.0), which means that the manuscript, images, and Supporting Information files will be freely available online, and any third party is permitted to access, download, copy, distribute, and use these materials in any way, even commercially, with proper attribution. For more information, see our copyright guidelines: http://journals.plos.org/plosone/s/licenses-and-copyright.

1. You may seek permission from the original copyright holder of Figure (1) to publish the content specifically under the CC BY 4.0 license. 

We replaced the figure which contained Open Street Map content by a new figure integrating BD ORTHO data from IGN (French National Geographic Institute), that are under a licence ‘Etalab 2.0’. We contacted IGN by email, asking for the right to use the illustration (that was sent attached to the mail) under a licence CC BY 4.0. We were answered that Etalab 2.0 licence is entirely compatible with CC BY 4.0 licence, and that BD ORTHO illustration made with BD ORTHO data can be distributed with a CC BY 4.0 licence as long as IGN and years are mentioned on the picture. Therefore, we modified the figure and added ‘IGN’ mentions and the ‘year’ on each panel. A copy of the conversation (in French) was added at the end of this document. We can also provide the pdf of this conversation.

3. PLOS journals require authors to make all data underlying the findings described intheir manuscript fully available without restriction unless the data are subject to ethical restrictions or owned by someone other than the authors (https://journals.plos.org/plosone/s/data-availability#loc-acceptable-data-access-restrictions). Therefore, we ask that you please upload underlying data to an appropriate data repository and update your Data Availability Statement accordingly or provide all contact details for where an interested researcher would need to apply to gain access to the relevant data. Please note that it is not acceptable for an author to be the sole named individual responsible for ensuring data access.

Following PLOS ONE requirements, all the data underlying the statistical analyses were made available in a figshare repository. They will be available if the manuscript would be considered for publication (doi: 10.6084/m9.figshare.20173217) in PLOS ONE and a link toward the manuscript will be done. 

4. In your Methods section, please provide additional information regarding the permits you obtained for the work. Please ensure you have included the full name of the authority that approved the field site access and, if no permits were required, a brief statement explaining why.

The following sentences were added at the end of the first section of Material and Methods: ‘Field site access was approved by the water managers of each lake: Communauté de Communes du Lac d'Aiguebelette (CCLA) for lake Aiguebelette, Syndicat Mixte d'Aménagement et de Développement de Serre-Ponçon (SMADESEP) and by Conseil Départemental des Bouches-du-Rhône for lake Etang des Aulnes. No specific authorization were necessary to filter water.’

5. In your Methods section, please include a comment about the state of the animals following this research. Were they released, euthanized or housed for use in further research? If any animals were sacrificed by the authors, please include the method of euthanasia and describe any efforts that were undertaken to reduce animal suffering

The paragraph concerning the gillnets data could have been confusing. Indeed, these fish data are collected by French water authorities, such the Water Agencies and the French Agency for the Biodiversity as part of the monitoring undergo for the Water Framework Directive (WFD). As authors we only have accessed to these data collected in national database, we did not sampled fish nor manipulated them. Our sampling design only concerned the filtration of water to collect eDNA.

This paragraph was modified to make it clearer: ‘We used data obtained from national fishing surveys conducted by the French Biodiversity Agency (OFB) and water agencies as part of the WFD monitoring programme. Fish data were collected according to CEN standard [51].’

6. We note that you have indicated that data from this study are available upon request. PLOS only allows data to be available upon request if there are legal or ethical restrictions on sharing data publicly. For more information on unacceptable data access restrictions, please see http://journals.plos.org/plosone/s/data-availability#loc-unacceptable-data-access-restrictions. 

As already mentioned, data supporting all the statistical analyses are now uploaded on figshare and have a DOI (10.6084/m9.figshare.20173217). They will be completely available without any request, if this manuscript is published. Moreover, in the metadata we will make a link pointing toward the paper URL.

Additional Editor Comments (if provided):

I got the recommendations and comments from an expert reviewer in the field. The reviewer agreed that the manuscript is technically sound and the data support the conclusions.

However, the lack of an explanation in the Methods and Results sections was suggested, and I share the comments. Therefore, I can invite you to submit a revised version of the manuscript that addresses the points raised by the reviewers.

Reviewers' comments:

Reviewer's Responses to Questions

Comments to the Author

1. Is the manuscript technically sound, and do the data support the conclusions?

Reviewer #1: Yes

2. Has the statistical analysis been performed appropriately and rigorously? 

Reviewer #1: Yes

3. Have the authors made all data underlying the findings in their manuscript fully available?

Reviewer #1: Yes

4. Is the manuscript presented in an intelligible fashion and written in standard English?

Reviewer #1: Yes

5. Review Comments to the Author

Reviewer #1: Comments:

In this study, the authors used eDNA and assessed the spatio-temporal variability of its signal for fish monitoring in lakes.

I suggest publishing the paper after some revisions.

L27: May authors change to "6 weeks" instead of "2 months" (according to Line 154)

On line 27, ‘2 months’ was changed by ‘6 weeks’: ‘oligotrophic lake) every 6 weeks for 1 year’

L32: maybe authors present here the traditional method used for the comparison.

This precision was added in parenthesis on lines 32-33: ‘traditional fish monitoring methods (standardized gillnet samplings)’

L77: May authors change fish to eukaryotic species

This was done on line 77: bacteria to eukaryotic species

L84: change to "The number of species that can be detected"

This was done on line 84: ‘The number of species that can be detected with eDNA’

L95: "focused" instead of "focussed"

‘focussed’ was changed by ‘focused’on lines 95-96: ‘Some studies of lacustrine fish focused on these questions’

L97: "both of the two aspects" instead of "the two apsects together"

This was done on line 97: ‘they consider both of the two aspects.’

L101-102: change to "considering possible changes over a year"

This was done one lines 101-102: ‘of the spatio-temporal variability of eDNA for several types of lakes and considering possible changes over a year’

L127: The subfigures may be named as (a) for Aiguebelette, (b) for Serre-Ponçon, and (c) Etang des Aulnes, instead of 1, 2, 3 which is a bit confusing

The subfigure names and the figure caption have been modified accordingly.

L235-241: This paragraph is results and not material and methods

We agree that this paragraph could have been confusing. As it was written, this paragraph looked like a result of our study, while this was an already known issue of the teleo primers, that was identified during the construction of the reference database (Valentini et al. 2016). The fact that some species could not be differentiated at the species level but at the genus levels was therefore part of the material and method and not of the result section. To avoid this confusion the paragraph was reworded. The last sentence of this paragraph concerning the marine species was moved to the result section (lines 309-311).

lines 242-246: ‘By using teleo primers, the following species are not differentiated [29] and are referred to as the genera: Salvelinus sp., Cottus sp. and Gobio sp. This is also the case for the allochthone acclimated species of Carassius genus. The teleo barcode does not discriminate some species belonging to different genera. They are grouped into ‘complexes’ [26] (see Supplementary Materials).’

L276: The thermal profile of Aiguebelette lake is missing from the Supplementary material.

The thermal profile of Aiguebelette lake was added to the Supplementary material

L279, 285, 340: Define the specific Supplementary Material (eg 1, 2..) in each case

‘Supplementary Material’ was replaced by the specific citation everywhere in the text, line 156 (S1 Table), lines 244-245 (see S2 Text), line 283 (S3 Fig), line 289 (S1 Table), line 348 (but see S4 Fig for the reads), line 404 (all figures in S5 Figs).

L293-295: This paragraph is a bit confusing. Someone cannot understand immediately why authors refer to Carassius carassius and Abramis brama. The first one is an allocthonous species and the reference in the second species is explained in the next sentece, however someone should know that the common bream is Abramis brama. Please be more clear in this paragraph.

We agree that this sentence was not clear, we therefore reworded it (lines 299-300): ‘Surprisingly, although 49 individuals of A. brama were captured in 2011, none was found in 2015, excluding common bream from the taxonomic inventory.’

Line 344-345: A comment about what the study of each site reveals (i.e. Fig 4), is missing

The goal pursues by the analysis on all the samples realised, was first to see if there was a distinction of these samples according to the lakes. As the lake was clearly the first factor differentiating the samples (as revealed by the non-overlapping of samples from two different lakes), we considered that the the spatio-temporal dynamics of the eDNA signal should be analysed lake by lake. This is what we tried to express in this paragraph: ‘The NMDS based on all eDNA samples (Bray–Curtis distance, stress = 0.156) showed that the fish fauna of the three sites were significantly different whatever the sampling location and period (PERMANOVA, p < 0.001; Fig. 4) as revealed by the non-overlapping point clouds. Due to these differences, the spatio-temporal dynamics of the eDNA signal was studied for each site independently.’ We think that the results of Fig 4 is detailed enough here.

2 more general comments:

1. In eDNA studies it is common to use technical repetitions for the samples. In this study, the authors filtered a big volume of water and further they evaluated all the results from all the sampling campaigns in each lake. Therefore, I understand why they chose this specific approach. However, authors may explain a bit more why they followed this sampling strategy.

The replication level may be reached through multiple ways (multiple samples per localities, multiple extractions per sample and multiple PCRs per extraction), as demonstrated by Ficetola [1]. We decided that instead of making 30 replicates of 1L per sampling location to use an integrative sampling strategy of 30L in a single filter cartridge, which limits the possibility to sample again exactly at the same place and possibility of 'contamination' between the field replicates. This volume is similar to the total volume of water collected with the single point sampling from other authors to collect the majority of the fish biodiversity [2–4]. In line 161 we explained why we preferred the integrative sampling. ‘This integrative sampling allowed to sample a large variety of habitats and to reduce the species detection biases due to local variation’. We privileged to increase the replication level in the laboratory steps, indeed we performed 12 PCR replicates to be sequenced, as suggested by [1] for rare species detection, and we increased the sequencing depth as it was demonstrated that the number of detected species increase with the sampling depth [5].

1. Ficetola GF, Pansu J, Bonin A, Coissac E, Giguet-Covex C, Barba M, et al. Replication levels, false presences and the estimation of the presence/absence from eDNA metabarcoding data. 2015;15. doi:10.1111/1755-0998.12338

2. Handley LL, Read DS, Winfield IJ, Kimbell H, Johnson H, Li J, et al. Temporal and spatial variation in distribution of fish environmental DNA in England’s largest lake. Environ Dna. 2019;1: 26–39. doi:10.1002/edn3.5

3. Macher T-H, Schütz R, Arle J, Beermann AJ, Koschorreck J, Leese F, et al. Beyond fish eDNA metabarcoding: Field replicates disproportionately improve the detection of stream associated vertebrate species. Metabarcoding Metagenomics. 2021;5: e66557. doi:10.3897/mbmg.5.66557

4. Bylemans J, Gleeson DM, Hardy CM, Furlan E. Toward an ecoregion scale evaluation of eDNA metabarcoding primers: A case study for the freshwater fish biodiversity of the Murray–Darling Basin (Australia). Ecol Evol. 2018. doi:10.1002/ece3.4387

5. Bylemans J, Gleeson DM, Lintermans M, Hardy CM, Beitzel M, Gilligan DM, et al. Monitoring riverine fish communities through eDNA metabarcoding: determining optimal sampling strategies along an altitudinal and biodiversity gradient. Metabarcoding Metagenomics. 2018;2: 1–12. doi:10.3897/mbmg.2.30457

2. It is proved in many eDNA studies that the temperature affects the eDNA. In this study it seems that it is not the case. However, I think that a discussion about this issue is missing.

In this study, we did not explicitly focused on this parameter or test it, but we integrate it in many aspects of the study and of the discussion. First, we provided vertical profile of temperature, to know if lakes were stratified or not, which was shown by Littlefair et al. to play a role on eDNA distribution in lakes. Moreover, when conducting a study over a year, temperature is expected to play an important role on lake functioning (stratification dynamics), on organism’s activities, phenology, and thus on eDNA production and decay. Here, we pointed out the link between fish reproduction periods and eDNA signal; however the design of the study does not allow to fully integrate temperature as an explanatory factor for our eDNA data. Adding a specific discussion on the effects of temperature on eDNA should be a little bit far from the topic of the manuscript, which is not dealing with the relative importance of the different factors affecting eDNA signal. Even if temperature have played a role, it would be difficult to specifically address all the possible direct and indirect effects of temperature on eDNA signal, the data available here do not offer the possibility to quantify these effects. Therefore, even if we recognize the interest of such topic, we cannot add a specific paragraph since we have not the relevant data to address this question; we are more focused on the general question of when and where to sample, in the scope of using eDNA in routine in monitoring programs.

6. PLOS authors have the option to publish the peer review history of their article (what does this mean?). If published, this will include your full peer review and any attached files.

Do you want your identity to be public for this peer review? For information about this choice, including consent withdrawal, please see our Privacy Policy.

Reviewer #1: No

De: contact.geoservices <contact.geoservices@ign.fr>

Envoyé: vendredi 24 juin 2022 09:47

À: Maxime LOGEZ

Objet: RE : Faire état d’une autre difficulté CRM:0107874

Bonjour,

Il n'y a aucune contre-indications dans les termes de la licence Etalab 2.0 , tant que vous indiquez la paternité et la date de dernière mise à jour de les ressources utilisées comme cela est fait sur votre document.

Cordialement,

Relation Utilisateurs - Géoservices IGN

Courriel : contact.geoservices@ign.fr

Afin de mieux répondre à vos attentes, nous vous invitons à partager votre avis sur notre nouveau site Géoservices

Les modalités d'accès aux services web évoluent au 1er février 2022

Cliquez ici pour en savoir plus

Inscrivez-vous à la lettre Géoservices (accessible en bas de page du site)

------------------- Message d'origine -------------------

De : pgie.geoservices.p <pgie.geoservices.p@agriculture.gouv.fr>;

Reçu : Thu Jun 23 2022 17:47:10 GMT+0200 (heure d’été d’Europe centrale)

À : contact.geoservices <contact.geoservices@ign.fr>;

Sujet : Faire état d’une autre difficulté

Informations sur le demandeur :

- Nom : Logez

- Prénom : Maxime

- Organisme : INRAE

- Adresse email : maxime.logez@inrae.fr

- Numéro de téléphone :

Description de la demande :

Bonjour, Je voulais savoir si la licence etalab 2.0 était compatible avec la licence CC BY 4.0. Je souhaite publier un article dans une revue en open access (Plos One) qui distribue son contenu sous licence cc BY 4.0. J'ai créer une illustration à partir de données IGN (BD Ortho, voir figure ci-jointe), en spécifiant l'origine des données. Est-ce que je peux la diffuser dans ce cadre (CC BY 4.0) ? Bien cordialement, Maxime Logez

Envoyé par Géoservices.ign.fr

Contactez-nous

Institut national de l'information géographique et forestère (IGN)

73 avenue de Paris 94165 SAINT-MANDÉ CEDEX

---

## [Editor Report · Decision Letter 1]

25 Jul 2022

Spatio-temporal variability of eDNA signal and its implication for fish monitoring in lakes

PONE-D-22-05624R1

Dear Dr. Logez,

We’re pleased to inform you that your manuscript has been judged scientifically suitable for publication and will be formally accepted for publication once it meets all outstanding technical requirements.

Kind regards,

Hideyuki Doi

Academic Editor

PLOS ONE

Additional Editor Comments (optional):

I carefully checked the revised manuscript as well as the response letter. I agree with the revisions and now can recommend publishing the paper.
---

## [Editor Report · Acceptance letter]

4 Aug 2022

PONE-D-22-05624R1 

Spatio-temporal variability of eDNA signal and its implication for fish monitoring in lakes 

Dear Dr. Logez:

I'm pleased to inform you that your manuscript has been deemed suitable for publication in PLOS ONE. Congratulations! Your manuscript is now with our production department. 

Kind regards, 

on behalf of

Dr. Hideyuki Doi 

Academic Editor

PLOS ONE